# The Preparation of Crumpled Graphene Oxide Balls and Research in Tribological Properties

**DOI:** 10.3390/ma17102383

**Published:** 2024-05-16

**Authors:** Lili Zhang, Zhengrui Zhang, Xi’an Gao, Hao Liao

**Affiliations:** College of Civil Engineering, Sichuan University of Science & Engineering, Zigong 643000, China; zhanglili@suse.edu.cn (L.Z.); gaoxian@suse.edu.cn (X.G.); 322085607101@stu.suse.edu.cn (H.L.)

**Keywords:** crumpled graphene oxide balls, preparation, characterization, tribological properties

## Abstract

In this study, crumpled graphene oxide balls (CGBs) were prepared via capillary compression using a rapidly evaporating aerosol droplet method. The CGBs were observed using scanning electron microscopy (SEM), high-resolution transmission electron microscopy (HRTEM), and Raman spectroscopy. The size distributions of crumpled particles were obtained using a laser nanometer particle size analyzer (DLS). The dispersibility of the water and the ionic liquid (IL) was tested by ultrasonic dispersion. The tribological properties of water or ionic liquids containing crumpled graphene oxide ball additives (W/IL-CGB) were tested by a reciprocating friction tester and compared with water/ionic liquids with graphene oxide. The morphology of the wear scar was observed by a three-dimensional optical microscope and its lubrication mechanism was analyzed. The results show that the CGBs were successfully prepared by rapid evaporation of aerosol droplets, and the obtained CGBs were crumpled paper spheres. The CGBs had good water dispersion and ionic liquid dispersion, and IL-CGB has excellent anti-friction and anti-wear effects on steel–steel friction pairs. During the friction process, the CGB was adsorbed at the interface of the steel–steel friction pair to form a protective layer, which avoids the direct contact of the friction pair, thereby reducing friction and wear.

## 1. Introduction

Ionic liquids (ILs) are low-melting temperature molten salts, typically comprising bulky anions and cations, that have low volatility, are nonflammable, have low melting points, possess high thermal stability, and exhibit a liquid phase over a broad temperature range [1]. Therefore, ionic liquids are suitable for use as lubricants. However, like other liquid lubricants, ILs exhibit insufficient load-carrying capacity and anti-wear performance at the boundary-lubrication state, which restrict their applications. In order to ensure effective lubrication, some additives are usually added to improve the performance of the lubrication system. We previously used graphene oxide (GO) as an oil additive and found that graphene oxide layers can effectively improve the performance of ILs in high-load conditions [2,3].

Graphene oxide has a unique two-dimensional layered structure, good mechanical properties, chemical stability, excellent anti-wear and anti-friction properties, and is a high-performance environmentally friendly lubricating additive [4,5,6,7,8,9,10,11,12]. However, due to its unique molecular structure, GO exhibits irreversible agglomeration in aqueous or oily media [2], which greatly reduces the lubrication performance of graphene and hinders its application and development in the field of lubrication. Therefore, it is of practical significance to control the microstructure or functional modification of graphene oxide to improve its dispersibility in aqueous or oily media. A crumpled graphene structure is a good choice for avoiding agglomeration [13].

Dou et al. prepared crumpled graphene balls as a high-performance additive that can significantly improve the lubrication properties of polyalphaolefin base oil [14]. In contrast to other carbon additives, whose tribological properties are sensitive to their concentration, crumpled graphene balls deliver consistently good performance between 0.01 wt% and 0.1 wt% concentration. It was found that crumpled graphene balls reduce the friction coefficient and wear coefficient by about 20% and 85%, respectively, with respect to the base oil.

In order to further improve the friction properties of CGB in lubricating oil, Zhang et al. loaded nano-magnesium silicate hydroxide onto the surface of CGB (MSH/CGB) [15]. Furthermore, in order to improve the compatibility with the base oil, the MSH/CGBs were decorated with oleic acid and stearic acid to obtain modified lipophilic composites (ML-MSH/CGB). Compared with base oil, the average friction coefficient, wearing capacity, and wear scar diameter of the as-prepared MSH/CGB oil were reduced by 25.4%, 22.1%, and 16.7%, respectively. 

In recent years, researchers have carried out a series of research works on the preparation of crumpled graphene balls (CGBs). The research involved the preparation of CGBs and the application of CGBs [13]. The preparation of CGBs mainly includes the thermal reduction method, aerosol spray method, mechanical method, and hydrothermal method.

Among these, the aerosol spray method is used to atomize the graphene oxide water dispersion into a small water droplet shape and uses the capillary force during water evaporation to make the graphene become a rough sphere [13]. This method starts with an aqueous solution, which is simple and environmentally friendly, has a fast processing time, and is suitable for continuous production. Researchers have explored several methods to avoid aggregation in solution, including reducing lamellar size, solvent–graphene interaction modification, and dispersants. Due to the simple preparation method, researchers also add other metal or metal oxide particles at the same time. In the process of graphene forming wrinkled graphene, other nanoparticles attach to the surface or pores of wrinkled graphene to form a composite structure. Larissa et al. prepared MnFe_2_O_4_-modified wrinkled graphene spheres using a one-step method. The shape of this wrinkled paper ball can be adjusted by adjusting the quantity of manganese ferrite nanoparticles added [16]. They explored two different applications of the obtained materials: electrochemical sensors for hydrogen peroxide and electrochemical supercapacitors.

The crumpled graphene spheres prepared by the above methods are used in many fields. Due to the microstructure of the crumpled graphene spheres, they have the characteristics of difficult stacking and more voids. Researchers loaded other substances on the CGBs to improve their application performance. Folded graphene spheres are mainly used in the applications of biosensors, gas sensors, chemical detection, and electrode preparation [17,18,19,20,21,22,23,24,25,26,27,28,29,30,31,32,33,34,35,36,37,38,39].

Therefore, changing the microstructure of GO may further improve the tribological properties of GO. In the present study, we changed the structure of lamellar graphene oxide into a graphene sphere by using the capillary force during water evaporation. The dispersibility of the water and the ionic liquid was compared by ultrasonic dispersion. The tribological properties of water or ionic liquids containing crumpled graphene oxide ball additives (W/IL-CGBs) were tested.

## 2. Materials and Methods

### 2.1. Material

Previous studies used ionic liquid 1-butyl-3-methylimidazolium tetrafluoroborate (purity, 97%), so the same ionic liquid was used in this experiment. ILs were synthesized and provided by the State Key Laboratory of Solid Lubrication, Lanzhou Institute of Chemical Physics. Powders of multilayer GO were purchased from Suzhou Tanfeng Graphene Technology Co., Ltd. (Suzhou, China). The water used in this experiment was ultrapure water. The Millipore Teflon filter was purchased from Longjin Membrane Technology Co., Ltd. (Nantong, China), diameter: 60 mm, pore size: 0.22 μm. Commercially available steel balls (AISI-52100) with 6 mm diameter and steel substrate were used for friction tests. The steel balls and substrate were ultrasonically cleaned in pure alcohol for each test. Other materials were used as received.

### 2.2. Preparation of CGB

The graphene oxide sheets were compressed into crumpled graphene balls using an aerosol capillary approach [35]. Following a typical synthesis procedure, an aqueous dispersion of micrometer-sized graphene oxide sheets (2.0 mg/mL) was nebulized to generate aerosol droplets (402AI, Jiangsu Yuyue Medical Equipment Co., Ltd., Zhenjiang, China) and flown through a tube furnace preheated at 400 °C. Rapid evaporation causes shrinkage of the droplets, first concentrating the graphene oxide sheets and subsequently compressing them into crumpled balls having a sub-micrometer scale. The products were filtered with a filter and dried at room temperature for 5 h, then placed into a vacuum dryer at 60 °C for 24 h. Figure 1 shows the experimental device and the schematic diagram. The temperature at both ends of the quartz tube is lower than that in the middle, so most of the CGB particles are attached to both ends of the inner wall of the quartz tube, and there is little CGB in the middle of the quartz tube. The GO heated for a short time can be collected by a silicone shovel at the front end of the quartz tube, as shown in Figure 1 Product 1 is named CGB-1; the GO heat-treated for a long time can be collected at the end of the quartz tube, as shown in product 2 (Figure 1), which is named CGB-2.

### 2.3. Micro-Morphology and Dispersion Stability Test of GO and CGB

The graphene oxide and CGB were characterized via high-resolution transmission electron microscopy (HRTEM, Talos F200S G2), scanning electron microscopy (SEM, Sigma300, Carl Zeiss Microscopy Limited, Shanghai, China), EDS spectra, and Raman spectroscopy (Thermo Fisher Scientific Dxr2xi, with 532 nm laser excitation, Waltham, MA, USA). The size distributions of crumpled particles were obtained by a laser nanometer particle size analyzer (DLS, Anton Paar Litesizer 500).

The same amounts (10 mg) of GO, CGB-1, and CGB-2 were added to three 5 mL glass bottles, and then 4 mL water was added to obtain the water dispersion system of the sample. The dispersion stability of different systems was investigated by ultrasonic dispersion for 30 min, power at 160 W, and static state comparison for 20 h and 96 h. The dispersion stability test in ionic liquids was the same as that in water.

### 2.4. Testing Methods

All the tribological performance experiments in this test were carried out on the comprehensive tester of material surface performance, as shown in Figure 2. In this experiment, an appropriate amount of steel sheet was first fixed on the bearing platform of the tester, and then the steel balls with a diameter of 6 mm were fixed by a clamp, and an aqueous solution containing GO, crumpled graphene (CGB), and an ionic liquid were added to the steel sheet. With a load of 500 g, the steel sheet was slowly dropped and clamped.

The experiment was compared with pure deionized water and ionic liquid. The three-dimensional optical fiber mirror was used to observe the surface morphology of the wear marks in this friction experiment, and the wear volume was measured. Each test was repeated three times to take the average value. The wear rate was calculated by the formula K = V/LS, where k is the wear rate, V is the wear volume, L is the load, and S is the wear stroke.

## 3. Results

### 3.1. Structural Analysis of CGB

The microstructures of GO and CGB were tested by SEM and HRTEM. The basic morphology is shown in Figure 2. GO has an obvious lamellar structure due to its thinness, and GO sheets have wrinkles similar to silk. The corresponding selected area’s electron diffraction pattern (in Figure 2c) of the graphene nanosheets shows an obvious six-fold pattern, which indicates the crystalline structure of the graphene [2].

Figure 3 shows the SEM morphology of the CGB; Figure 3a,b is CGB-1 and Figure 3c,d is CGB-2. Figure 3b,d show the enlarged graphs of Figure 3a and Figure 3c, respectively. The structure of CGB is similar to that of crumpled paper balls. In Figure 3a, most of the crumpled balls can be seen, but a small part is still in the lamellar layer. In Figure 3c, almost all of the GO has been crumpled. This is because CGB-1 is a product collected at the front end of the tube furnace, and the heat treatment process is incomplete, so the GO part is not completely shrunk. CGB-2 has a long heating process, so it shrinks completely. The wrinkle degree of CGB-1 is significantly lower than that of CGB-2 after heat treatment.

In addition, the morphology of CGB was observed by HRTEM, as shown in Figure 4. Figure 4a shows CGB-1 and Figure 4b shows CGB-2. From the figure, it can also be seen that the microstructure of GO has changed after the heat treatment. When the heat treatment time is short, the shrinkage is incomplete, and part of the layer can still be seen. Figure 4c,d show the corresponding selected area electron diffraction patterns of CGB-1 and CGB-2. The diffraction pattern of CGB-1 can still be a six-fold pattern, but it is not as obvious as that of GO. The diffraction pattern of CGB-2 is irregular, which can also indicate that the microstructure of CGB-2 changes more than that of CGB-1. The percentage values of carbon and oxygen were obtained through EDS, with GO containing 70.89% C and 29.11% O (C/O = 2.44), CGB-1 containing 83.03% C and 16.97% O (C/O = 4.89), and CGB-2 containing 86.59% C and 13.41% O (C/O = 6.46). This shows that the structure of CGB does not change significantly after heating at 400 °C in a tube furnace, and the content of O atoms decreases gradually, which may be due to the decomposition of labile oxygen functional groups in the GO during heating.

The Raman spectra of GO and CGB deposited onto slide glass were recorded; Figure 5a. As can be seen in Figure 5a, all the spectra present D and G peaks centered at ~1350 cm^−1^ and ~1585 cm^−1^, respectively, and the second-order band centered at 2400–3300 cm^−1^. The sizes of the nanoparticles were tested by DLS, as shown in Figure 5b. With longer heating time, the average size of the CGB became smaller. The average size of CGB-1 was approximately 586 nm and the average size of CGB-2 was approximately 495 nm. Although the two particles are similar, it can be seen from Figure 5b that the distribution range of CGB-1 is wider than that of CGB-2, indicating that the longer the heating time, most of the GO shrinks with N_2_ flow. 

### 3.2. Dispersion Stability of GO and CGB

The wrinkled graphene spheres were dispersed in deionized water at a certain concentration, and the graphene oxide GO aqueous dispersion without any modification was used as the blank group control. The three samples were subjected to ultrasound. After 10 min of ultrasound, most of the graphene oxide was still agglomerated, and the latter two samples were completely dispersed in water. After 30 min of ultrasound, graphene oxide was almost completely dispersed. After standing for 20 h, it can be seen from Figure 6c that the color of the dispersion samples of CGB-1 and CGB-2 did not change significantly, and the GO sample had a small amount of precipitation. After standing for 96 h, it can be seen from Figure 6d that the GO aqueous dispersion was completely precipitated, most of CGB-1 was precipitated, and only a small part of CGB-2 was agglomerated. It can be seen that the wrinkled graphene spheres have good dispersion stability in water, which also shows that the wrinkled graphene spheres can effectively improve the dispersion performance of graphene in water.

To study the tribological properties of CGB in ionic liquids, the dispersion stability of CGB in ionic liquids was also compared. The results are shown in Figure 7. These three samples were subjected to ultrasound. After 10 min of ultrasound, most of the GO was still agglomerated, and the latter two samples were completely dispersed in the ionic liquid. After 20 min of ultrasound, GO dispersed most of it, and after 1 h of ultrasound, GO was almost completely dispersed. After standing for 2 days, it can be seen from Figure 7d that the color of the dispersion samples of CGB-1 and CGB-2 did not change significantly, and there was no obvious agglomeration. Only a small amount of black precipitate was found at the bottom of the reagent bottle. On the contrary, the GO sample as the blank control group had significant agglomeration.

### 3.3. Tribological Properties

Figure 8 shows the curves of the friction coefficient of water dispersion and ionic liquid dispersion containing three kinds of lubricating additives with time at a concentration of 2.5 mg/mL or 0.25 mg/mL, in which each group of curves has pure water (W-0) or pure ionic liquid (IL-0) as blank control. It can be seen from Figure 8a that the friction coefficient of W-GO and W-CGB is lower than that of W-0, so nano-additives can indeed reduce the friction coefficient, but it may be that the concentration of GO and CGB is higher and the difference is not significant. Figure 8b shows that the friction coefficient of W-CGB-1 is the lowest in the later stage of friction experiment; Figure 8c shows that the friction coefficient of W-CGB-1 is lower in the early stage of friction experiment; and Figure 8d shows that the friction coefficient of IL-CGB-1 and IL-CGB-2 is much lower than that of IL-0 and IL-GO in most of the friction experiments. Therefore, nano-additives can play a better role at the appropriate concentration. When the concentration is too high, such as 2.5 mg/mL, the nano-additives will hinder the friction. At 0.25 mg/mL, the effect of reducing friction and wear in the ionic liquid is better than in water. It can be seen from Figure 8b,d that when the concentration is 0.25 mg/mL, whether it is water or IL when CGB-1 is used as an additive, the friction coefficient is lower and more stable, so CGB-1 is better than CGB-2.

According to the results of the friction coefficient test, it can be seen that when the concentration of the nano-additive is 0.25 mg/mL, the ionic liquid has the best effect as a lubricant, so only the wear rate after the friction test is tested in Figure 8d. Figure 9 is a three-dimensional surface profile image of the wear marks under the IL with lubricating additives. It can be seen from Figure 9 that the wear scars become shallower in turn. Figure 10a is a two-dimensional graph of the wear scar. It can be seen from the figure that the ionic liquid (IL-0) without any additive has the largest wear area, followed by the ionic liquid IL-GO, and the IL-CGB-1 and IL-CGB-2 with crumpled graphene oxide balls have the smallest wear area. The possible reason for this is that the crumpled graphene oxide balls are adsorbed at the interface of the steel–steel friction pair during the friction process to form a micro-ball state, which achieves the effect of the protective layer and avoids the direct contact of the friction pair, thereby reducing the friction and wear. The wear rate of IL-0 can also be seen from the wear rate of the Figure 10b diagram, which shows the wear rate of IL-CGB-2 is the smallest. This may be due to the high degree of shrinkage of CGB-2 and the good isolation effect, so the anti-wear effect is good.

The micro-morphology of the worn surface was also observed. Figure 11 shows the SEM photos of the wear scar when ionic liquid dispersions containing different lubricating additives were used as a lubricating medium. It can be seen from Figure 11a that the wear scar width was 260.1 μm when the pure ionic liquid was lubricated, and the wear scar width was the largest. Figure 11b shows the width of the wear scar was 223.4 μm when the IL-GO was lubricated, and the width of the wear scar decreased, indicating that its anti-wear performance was improved, but this was not obvious. The wear scar widths of IL-CGB-1 and IL-CGB-2 in Figure 11c,d are 215.8 μm and 210.4 μm, respectively. The wear scar width is greatly reduced and its surface is smooth. It can be seen that the ‘filling’ and ‘isolation’ of CGB between the friction pairs reduce the surface wear. CGB as a lubricating additive significantly improves the anti-wear performance of the ionic liquid. Comparing CGB-1 and CGB-2, the surface of CGB-1 is smoother after the friction test. The wear rate of IL-CGB-1 is close to that of IL-CGB-2, so considering the friction coefficient and wear rate, the tribological performance of CGB-1 is better.

The microstructure of the GO and CGB in lubricants after friction testing was characterized via HRTEM. The lubricants on the wear track after friction testing were collected by a microsyringe, and diluted with water by ultrasonic dispersion for a long time. The suspension was then dropped on a lacy carbon-coated Cu grid for TEM observations. Figure 12 shows TEM images of the debris after friction testing. It can be clearly seen that GO has undergone significant changes after the friction test, as shown in Figure 12b. Compared with the initial GO, GO is significantly thicker and stacked. CGB can still maintain its original state after friction test. Therefore, by changing the microstructure of GO to a wrinkled shrinkage state, the effect of nano-additives can be prolonged.

### 3.4. Related Wear Mechanisms

Through the above tests, we summarized the friction and wear mechanism of CGB. Figure 13 shows a schematic of the mechanisms of the ILs with GO or CGB during friction. The addition of GO and CGB to the ionic liquid can further improve the tribological properties. Both fillers can be adsorbed on the surface of the steel to form a protective layer to avoid direct contact. Compared with GO, the microstructure of CGB is similar to spherical, and there is a rolling effect in the friction process, which changes the sliding friction into rolling friction, so the friction and wear are further reduced. At the same time, after a long-running friction test, graphene oxide will stack, and the relative concentration in the oil will decrease, while the dispersion stability of CGB is relatively good, so the anti-friction and anti-wear effect of CGB will be better than that of GO. In addition, CGB-1 with intermediate dimensions may be even better. CGB-1 has a crumpled structure, and also has lamellar parts, which wrinkled CGB-1 can avoid stacking after long-term friction, and lamellar parts form a protective layer between the counterparts.

## 4. Conclusions

Using capillary compression in rapidly evaporating aerosol droplet methods, crumpled graphene oxide balls were prepared. The degree of shrinkage of CGBs is different with different heating times. The dispersion stability of CGBs in water or ionic liquid was improved.The tribological properties of CGB nano-additives were studied. Compared with GO, tribological performance was further improved. CGB-1 retains a part of the lamellar structure and partially shrinks. During the friction process, a protective layer is formed on the surface of the friction pair. At the same time, the part that shrinks into a wrinkled ball can avoid stacking after long-term friction. Therefore, the effect of partially shrunk CGB-1 is better than that of CGB-2.To obtain the best lubricants with improved friction-reduction and wear-resistance properties in various lubricating states, fine-tuning of the degree of crumpling, as a function of the operating conditions (mainly temperature, concentration, and N_2_ flow rate), will be investigated in our future work.

## Figures and Tables

**Figure 1 materials-17-02383-f001:**
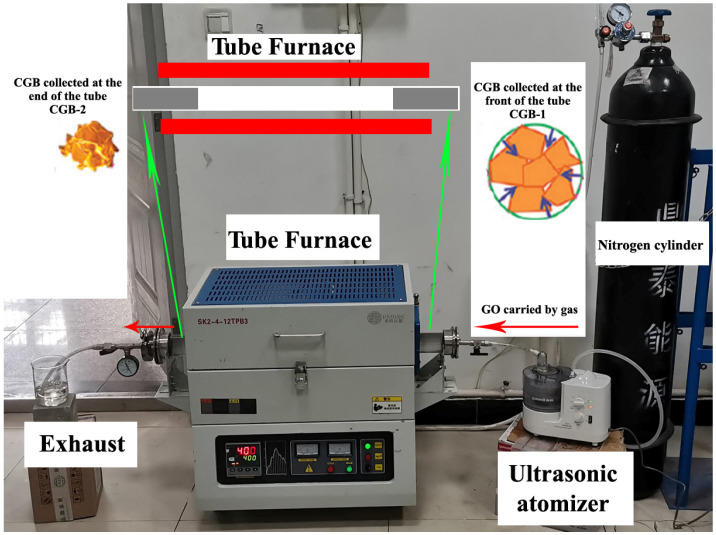
Experimental device and the schematic diagram.

**Figure 2 materials-17-02383-f002:**
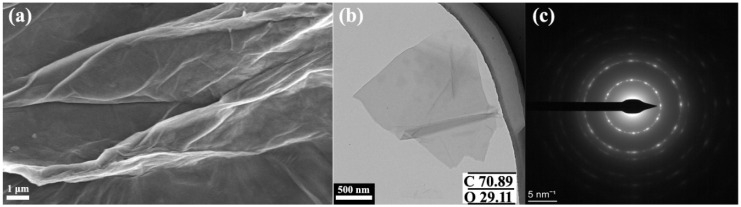
SEM and TEM images of GO. (**a**) SEM image of GO, (**b**) TEM image of GO, (**c**) corresponding SAED pattern of graphene oxide nanosheets. The inset (**b**) shows the percentage values of carbon and oxygen.

**Figure 3 materials-17-02383-f003:**
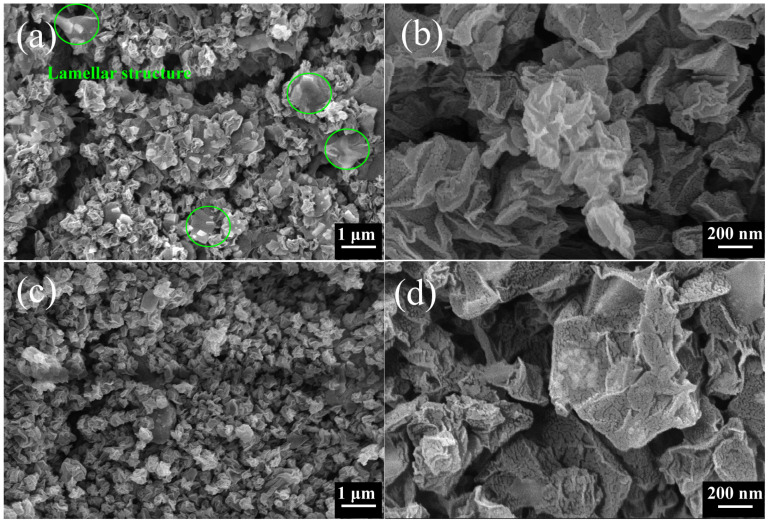
SEM images of CGB. (**a**) CGB-1, (**b**) partial enlargement of (**a**), (**c**) CGB-2, (**d**) partial enlargement of (**c**).

**Figure 4 materials-17-02383-f004:**
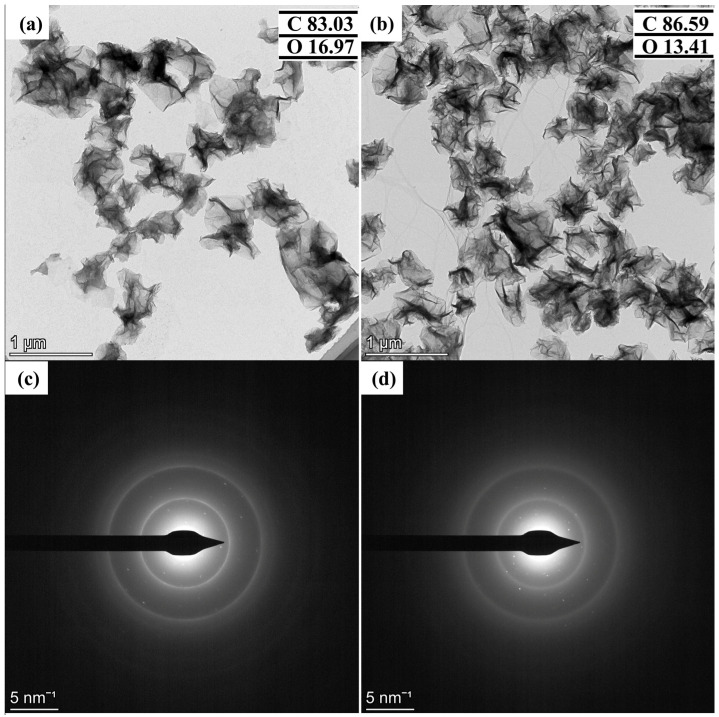
TEM images and the corresponding selected area electron diffraction pattern of CGB-1 and CGB-2. (**a**,**c**) CGB-1, (**b**,**d**) CGB-2. The insets of (**a**,**b**) are percentage values of carbon and oxygen.

**Figure 5 materials-17-02383-f005:**
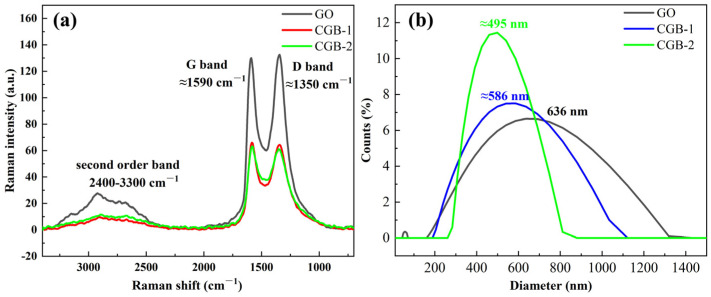
(**a**) Raman spectra of GO and CGB; (**b**) the size distribution of the GO and CGB nano-additives.

**Figure 6 materials-17-02383-f006:**
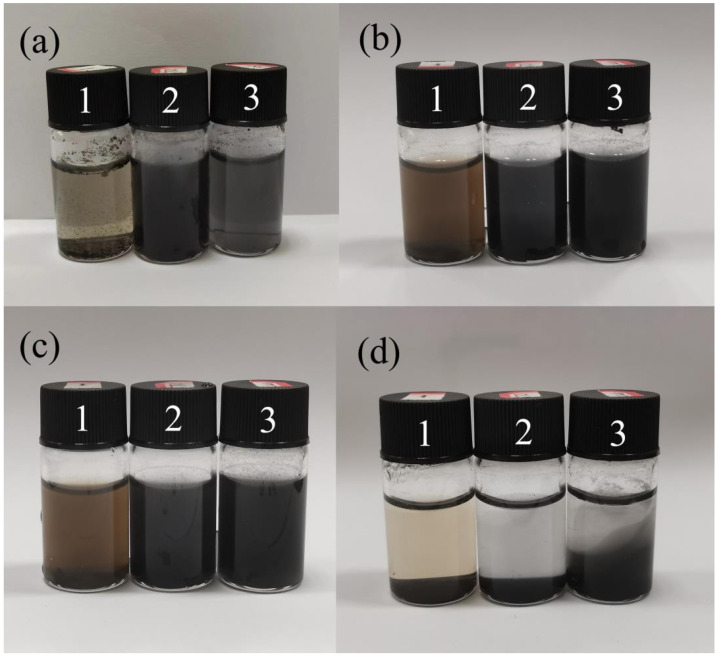
The dispersion results of 1—GO, 2—CBG-1, and 3—CBG02 in the water system. (**a**,**b**) The dispersion effects of ultrasound for 10 min and 30 min are represented, respectively. (**c**,**d**) The dispersion effects after standing for 20 h and 96 h are presented, respectively.

**Figure 7 materials-17-02383-f007:**
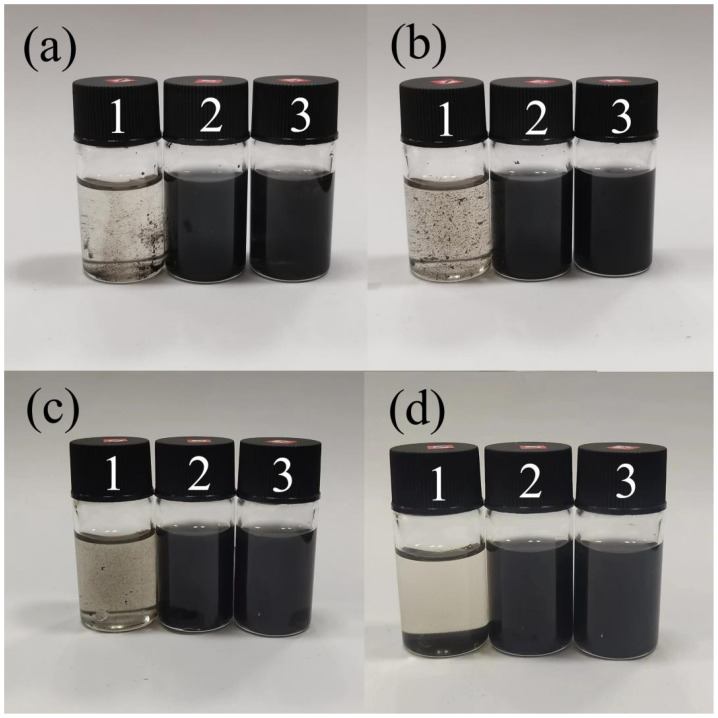
The dispersion results of 1—GO, 2—CBG-1, and 3—CBG-2 in the ionic liquid system. (**a**–**c**) Represent the dispersion effect of ultrasound for 10 min, 20 min, and 1 h, respectively. (**d**) Represents the dispersion effect after standing for 48 h.

**Figure 8 materials-17-02383-f008:**
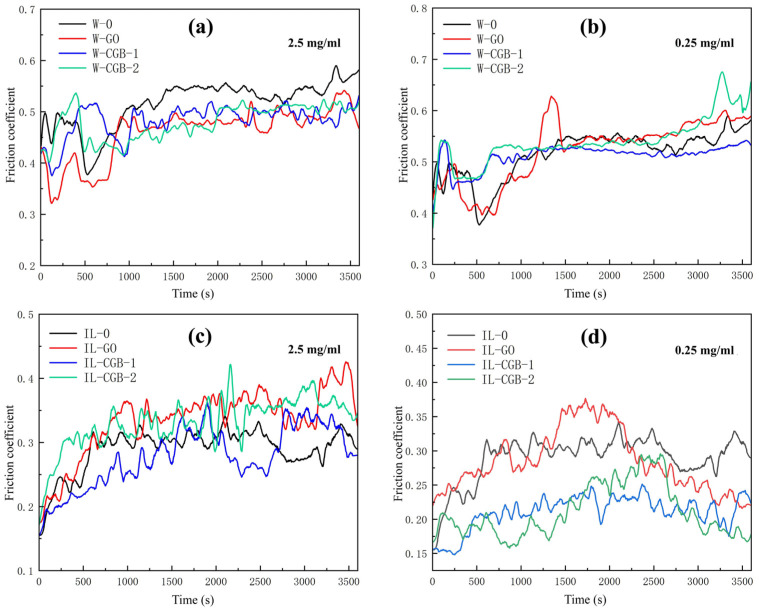
Friction coefficient–time curves of different concentrations of water dispersions and ionic liquid dispersions with different graphene oxide additives. (**a**) water, 2.5 mg/mL; (**b**) water, 0.25 mg/mL; (**c**) IL, 2.5 mg/mL; (**d**) IL, 0.25 mg/mL.

**Figure 9 materials-17-02383-f009:**
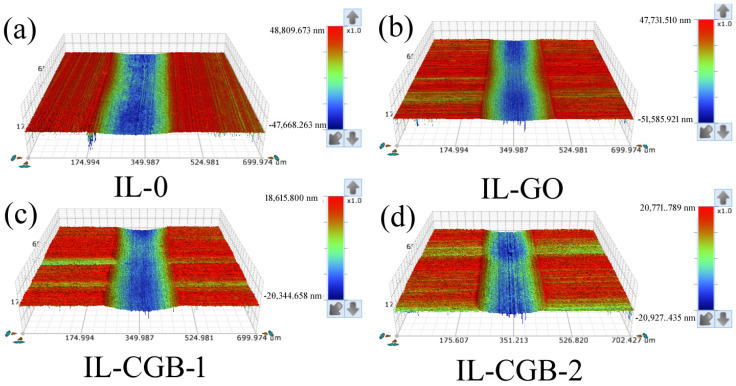
Three-dimensional profile of wear scars of IL with different lubricating additives.

**Figure 10 materials-17-02383-f010:**
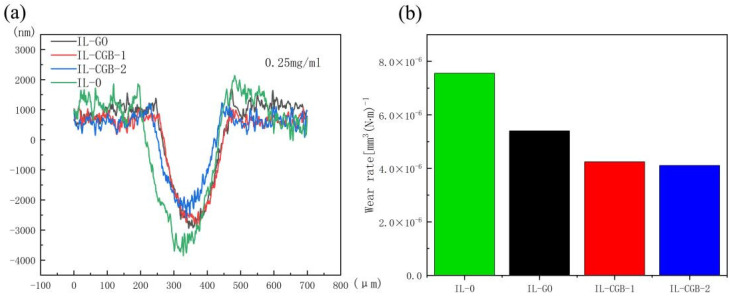
Two-dimensional curve and wear rate of wear scar under IL with different lubricating additives. (**a**) Two-dimensional curve of wear scar; (**b**) wear rate of wear scar.

**Figure 11 materials-17-02383-f011:**
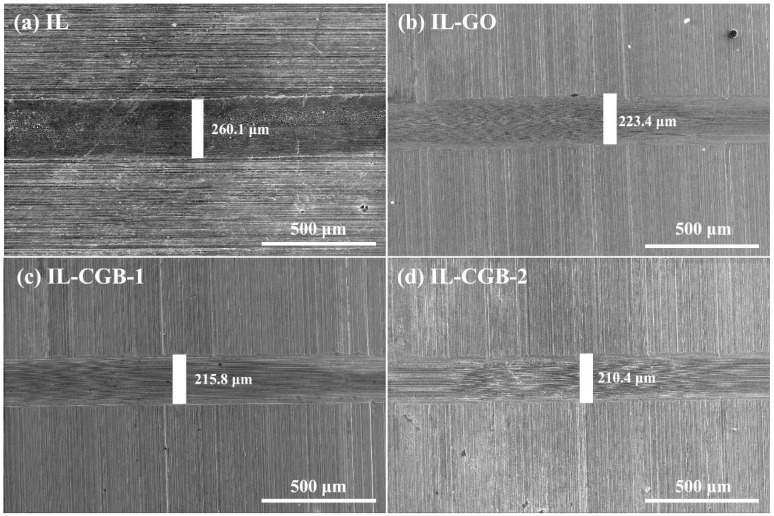
SEM images of the wear scar after the friction test. (**a**) IL, (**b**) IL-GO, (**c**) IL-CGB-1, (**d**) IL-CGB-2.

**Figure 12 materials-17-02383-f012:**
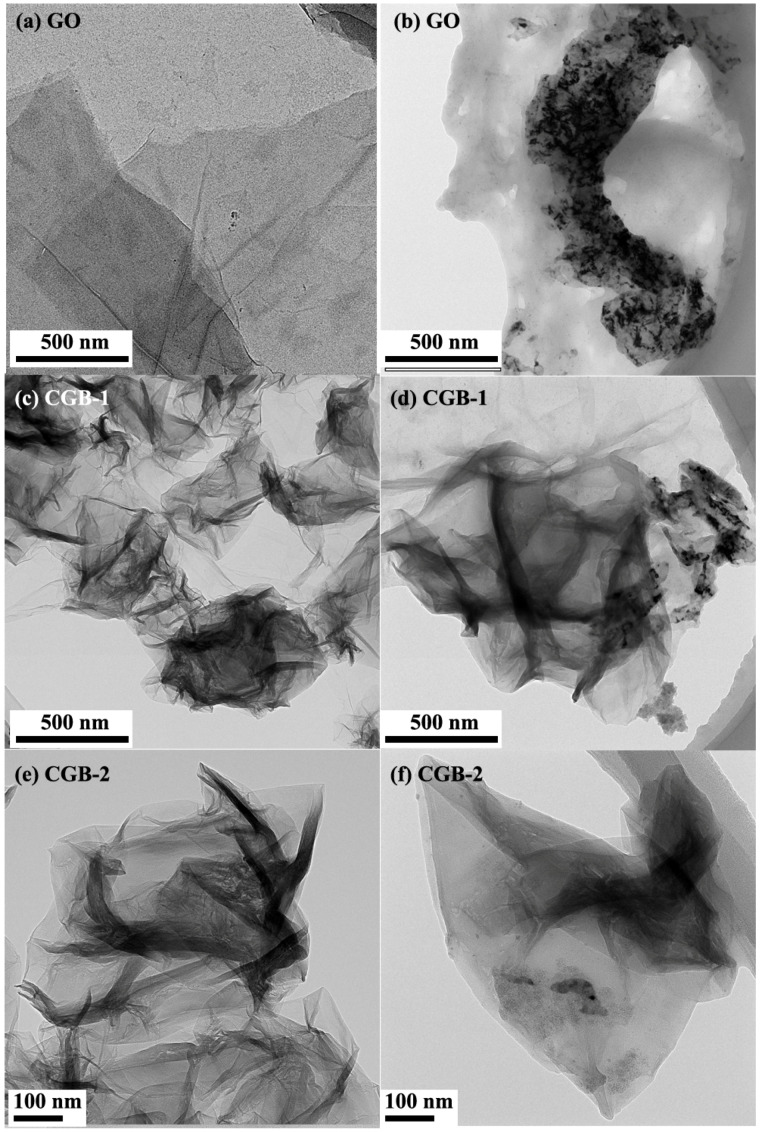
TEM images of GO, CGB, and the debris after the friction test: (**a**) GO; (**b**) GO after friction testing; (**c**) CGB-1; (**d**) CGB-1 after friction testing; (**e**) CGB-2; (**f**) CGB-2 after friction testing.

**Figure 13 materials-17-02383-f013:**
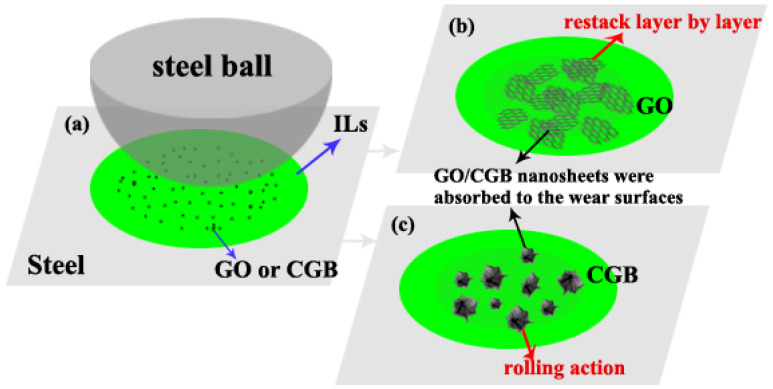
Schematic of the mechanisms of the IL with GO or CGB nano-additives during friction. (**a**) Schematic of the IL with GO or CGB during friction test; (**b**) GO restack after friction test; (**c**) CGB after friction test.

## Data Availability

The data used to support the findings of this study are available from the corresponding author upon request.

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
