# Peer review of "The Preparation of Crumpled Graphene Oxide Balls and Research in Tribological Properties"

_materials, 2024, doi:10.3390/ma17102383_

Round 1
Reviewer 1 Report
Comments and Suggestions for Authors
The authors have reported on tribological properties of crumpled graphene oxide, and its dispersion in ionic liquid. There are a few issues that need to be addressed before suggesting this paper for publication.
Many statements in the introductions are not supported with citation.
The authors should provide elemental analysis of the synthesized CGB, such as EDX and Raman to confirm the formation of graphene oxide.
The legends in the Fig.7 are not in English.
Some abbreviations such as FET should be stated what they stand for also more examples for biosensing applications of carbon-based materials should be included e.g doi.org/10.1016/j.snb.2020.128703
The captions of Figures are not clear and specific. The Authors should mention in the Figures’ caption what each panel is.
Author Response
Dear editor and reviewers,
Thanks a lot for your patient and constructive comments. We have carefully revised our manuscript by considering the reviewer’s comments. A point-by-point reply to the reviewer’s comments and revision details are as follows:
The authors have reported on tribological properties of crumpled graphene oxide, and its dispersion in ionic liquid. There are a few issues that need to be addressed before suggesting this paper for publication.
- Many statements in the introductions are not supported with citation.
Response:
The introduction paragraph and references have been re-edited. The corresponding modification is on page 1-2 of the article.
- The authors should provide elemental analysis of the synthesized CGB, such as EDX and Raman to confirm the formation of graphene oxide.
Response:
Both EDX and Raman of GO and CGB were tested. EDS test was added to Fig.2 and Fig.4. The percentage values of carbon and oxygen were obtained through EDS, with GO containing 70.89 % C and 29.11 % O (C/O = 2.44), CGB-1containing 83.03 % C and 16.97 % O (C/O = 4.89) and CGB-1containing 86.59 % C and 13.41 % O (C/O = 6.46). It shows that the structure of CGB does not change significantly after heating at 400 oC in tube furnace, and the content of O atom decreases gradually, which may be due to the decomposition of labile oxygen functional groups in the GO during heating.
Fig.2 SEM and TEM picture of GO. The inset (b) is percentage values of carbon and oxygen.
Fig. 4 TEM picture and the corresponding selected area electron diffraction pattern of CGB-1 and CGB-2. (a), (c) CGB-1, (b), (d) CGB-2. The inset (a) and (b) are percentage values of carbon and oxygen.
Raman spectra of GO and CGB deposited onto slide glass were recorded, Fig. 5(a) As can be seen in Fig 5(a), all the spectra present D and G peaks centered at ∼1350 cm−1 and ∼1585 cm−1 respectively, and the second-order band centered at 2400-3300 cm−1. The corresponding modification is on page 8 of the article.
Figure 5. (a) Raman spectra of GO and CGB; (b) the size distribution of the GO and CGB nano-additives.
The corresponding modification is on page 1-2 of the article.
- The legends in the Fig.7 are not in English.
Response:
Please forgive my carelessness, the Fig.7 has been modified as required. The following picture is the modified Figure 7(Figure 8 in the present paper). The corresponding modification is on page 10 of the article.
Figure 7. Friction coefficient-time curves of different concentrations of water dispersions and ionic liquid dispersions with different graphene additives.
- Some abbreviations such as FET should be stated what they stand for also more examples for biosensing applications of carbon-based materials should be included e.g doi.org/10.1016/j.snb.2020.128703.
Response:
FET means field-effect transistor. But in introduction paragraph section, relevant description was deleted, so it cannot be seen in the article. The corresponding modification is on page 2 of the article.
The relevant research was learned and cited in the research paper. The corresponding modification is on page 2 of the article.
[28] Keshavarz M.; Chowdhury A.; Kassanos P.; Tan B.; Venkatakrishnan K.; Self-assembled N-doped Q-dot carbon nanostructures as a SERS-active biosensor with selective therapeutic functionality. Sensors & Actuators: B. Chemical, 2020,323, 128703.
The captions of Figures are not clear and specific. The Authors should mention in the Figures’ caption what each panel is.
Response:
The unclear captions of figures are re-edited, the modifications are as follows:
Figure 2. SEM and TEM images of GO. (a) SEM image of GO, (b) TEM image of GO, (c) corresponding SAED pattern of graphene oxide nanosheets. The inset (b) is percentage values of carbon and oxygen.
Figure 3. SEM images of CGB. (a) CGB-1 (b) partial enlargement of (a), (c) CGB-2 (d) partial enlargement of (c).
Figure 4. TEM picture and the corresponding selected area electron diffraction pattern of CGB-1 and CGB-2. (a), (c) CGB-1, (b), (d) CGB-2. The inset (a) and (b) are percentage values of carbon and oxygen.
Figure 11. TEM images of GO, CGB and the debris after friction test: (a) GO; (b) GO after friction testing; (c) CGB-1; (d) CGB-1 after friction testing; (e) CGB-2; (f) CGB-2 after friction testing.

Reviewer 2 Report
Comments and Suggestions for Authors
Dear Sir,
The topic of this study is interesting, but there two main draw-backs: one would be the questionable quality of the English language used (which can be easily corrected and improved, in order to eliminate awkward expressions, typos an grammar errors) and te second would be the logic behind the Introduction paragraph.
In that aspect, the authors started by mentioning that “However, due to its unique molecular structure, graphene will exhibit irreversible agglomeration in aqueous or oily media, which greatly reduces the lubrication performance of graphene and hinders its application and development in the field of lubrication.”, an afirmation without any reference given, and continued by stating that it would be interesting to “… control the microstructure or functional modification of graphene oxide to improve its dispersibility in aqueous or oily media.” – which is the correct approach to tackle this problem. However, after a good start, instead of mentioning the benefits, from a triblogical point of view, of the use of crumbled graphene balls (e.g. Dou et al., Self-dispersed crumpled graphene balls in oil for friction and wear reduction, PNAS, 113 (6) 1528-1533, https://doi.org/10.1073/pnas.1520994113 or Zhang et al., Nano-Magnesium Silicate Hydroxide/Crumpled Graphene Balls Composites, a Novel Kind of Lubricating Additive with High Performance for Friction and Wear Reduction, Materials 2020, 13(17), 3669; https://doi.org/10.3390/ma13173669 and other similar papers), the authors mention numerous other uses of such a material, which are pertaining to the main topic of their study.
They should have presented in the Introduction paragraph the following:
1. how CGBs were previously used to improve tribological properties of some lubricants (mainly oils)
2. the reason for using ionic liquids (with previous examples) – were CGBs been previously used in a ionic liquid?
3. short presentation on the methods for obtaining the desired CGBs
And if they desire, a short paragraph on other uses of CGBs.
The preparation of CGB-1 and -2 needs some clarification on how exactly they were separated: according to the experimental protocol, an aqueous dispersion of micrometer-sized graphene oxide sheets was first obtained, this suspension was nebulized to generate aerosols and these were next flown through a tube furnace preheated at 400C. How exactly (and why) some particles were rapidly deposed at the entrance (forming the CGB-1 fraction), while other were carried all along the tube of the furnace and collected at the exit (forming the CGB-2 fraction). At 400C, water must have instantly evaporated, then graphen particles were transported along the tube by the carrier gas. At the end of the process, how exactly were the 2 fractions collected? Was there such a neat separation among CGB-1 and -2 that no third fraction (along the inside length of the tube) existed? Was there any attempt to evaluate the dimensions of the CGBs in the two fractions (in other words, could CGB-1 presents bigger diameters of the crumpled balls -and therefore have higher mass - than CGB-2?) – could this be directly related to the observed results (that CGB-1 is better than CGB-2)? Fig. 11, in which the magnification of both CGBs is different, could point toward different dimensions of the two CGBs.
Did the authors observed through SEM imaging the wearing on the balls used?
Otherwise, the experimental strategy is correct and the results interesting. Since GO as a ball does not function well due to aggregation and agglomeration, a shrinking toward a modified structure was attempted, but the authors’ results showed that too much shrinking/crumpling is not beneficial. That would open the question: which degree of crumpling is the best?
Provided that the authors modify their Introduction and address the above questions, the manuscript could be considered for publiction.
Comments on the Quality of English Language
The quality of the language used must be improved
Author Response
Dear editor and reviewers,
Thanks a lot for your patient and constructive comments. We have carefully revised our manuscript by considering the reviewer’s comments. A point-by-point reply to the reviewer’s comments and revision details are as follows (in order of the reviewer’s comments):
The topic of this study is interesting, but there two main draw-backs: one would be the questionable quality of the English language used (which can be easily corrected and improved, in order to eliminate awkward expressions, typos and grammar errors) and the second would be the logic behind the Introduction paragraph.
In that aspect, the authors started by mentioning that “However, due to its unique molecular structure, graphene will exhibit irreversible agglomeration in aqueous or oily media, which greatly reduces the lubrication performance of graphene and hinders its application and development in the field of lubrication.”, an affirmation without any reference given, and continued by stating that it would be interesting to “… control the microstructure or functional modification of graphene oxide to improve its dispersibility in aqueous or oily media.” – which is the correct approach to tackle this problem. However, after a good start, instead of mentioning the benefits, from a triblogical point of view, of the use of crumbled graphene balls (e.g. Dou et al., Self-dispersed crumpled graphene balls in oil for friction and wear reduction, PNAS, 113 (6) 1528-1533, https://doi.org/10.1073/pnas.1520994113 or Zhang et al., Nano-Magnesium Silicate Hydroxide/Crumpled Graphene Balls Composites, a Novel Kind of Lubricating Additive with High Performance for Friction and Wear Reduction, Materials 2020, 13(17), 3669; https://doi.org/10.3390/ma13173669 and other similar papers), the authors mention numerous other uses of such a material, which are not pertaining to the main topic of their study.
They should have presented in the Introduction paragraph the following:
- how CGBs were previously used to improve tribological properties of some lubricants (mainly oils)
- the reason for using ionic liquids (with previous examples) – were CGBs been previously used in an ionic liquid?
- short presentation on the methods for obtaining the desired CGBs
And if they desire, a short paragraph on other uses of CGBs.
Response:
We have carefully improved English for the revised manuscript. We got help from a native English speaker. The modified sections are highlighted in the revised manuscript.
Introduction paragraph has been re-edited. It contains four parts. (1) The reason for using ionic liquids (with previous examples) (2) GO and CGBs were previously used to improve tribological properties of some lubricants (mainly oils) (3) Presentation on the methods for obtaining the desired CGBs. (4) A short paragraph on other uses of CGBs.
The preparation of CGB-1 and -2 needs some clarification on how exactly they were separated: according to the experimental protocol, an aqueous dispersion of micrometer-sized graphene oxide sheets were first obtained, this suspension was nebulized to generate aerosols and these were next flown through a tube furnace preheated at 400 oC. How exactly (and why) some particles were rapidly deposed at the entrance (forming the CGB-1 fraction), while other were carried all along the tube of the furnace and collected at the exit (forming the CGB-2 fraction). At 400 oC, water must have instantly evaporated, then graphene particles were transported along the tube by the carrier gas. At the end of the process, how exactly were the 2 fractions collected? Was there such a neat separation among CGB-1 and -2 that no third fraction (along the inside length of the tube) existed? Was there any attempt to evaluate the dimensions of the CGBs in the two fractions (in other words, could CGB-1 presents bigger diameters of the crumpled balls -and therefore have higher mass - than CGB-2?) – could this be directly related to the observed results (that CGB-1 is better than CGB-2)? Fig. 11, in which the magnification of both CGBs is different, could point toward different dimensions of the two CGBs.
Response:
The temperature at both ends of the quartz tube is lower than that in the middle, so most of the CGB particles are attached to both ends of the inner wall of the quartz tube, there is little CGB in the middle of the quartz tube. The short time heated GO can be collected by silicone shovel at the front end of the quartz tube, as shown in Figure 1 (a). Product 1, is named CGB-1; the long time heat-treated GO can be collected at the end of the quartz tube, as shown in product 2 in Fig.1, which is named CGB-2. The relevant change has been added in page 3.
Fig.1 Experimental device and the schematic diagram.
The sizes of the nano-particles were tested by DLS, as shown in Figure 5(b). With longer heating time, the average size of the CGB became smaller. The average size of CGB-1 is approximately 495 nm, and the average size of CGB-2 is approximately 586 nm. Although the two particles are similar, it can be seen from Figure 5(b) that the distribution range of CGB-1 is wider than that of CGB-2, indicating that the longer the heating time, most of the GO shrinks with N2 flow.
Figure 5. (a) Raman spectra of GO and CGB; (b) the size distribution of the GO and CGB nano-additives.
In Figure 11, the lubricants on the wear track after friction testing were collected by microsyringe, and diluted with water by ultrasonic dispersion for a long time. IL-CGB on the wear track was very few, so little CGB was found by TEM test. TEM test in Fig 11 does not represent the size of all the CGB. Size distribution was tested by DLS.
Did the authors observed through SEM imaging the wearing on the balls used?
Response:
I am very sorry that SEM imaging the wearing on the balls was not observed. We just show the wear scar of the steel plate. We will pay attention to this part of the test in the next study.
Otherwise, the experimental strategy is correct and the results interesting. Since GO as a ball does not function well due to aggregation and agglomeration, a shrinking toward a modified structure was attempted, but the authors’ results showed that too much shrinking/crumpling is not beneficial. That would open the question: which degree of crumpling is the best?
Response:
In my opinion, like CGB-1 is suitable, according to the TEM images, CGB-1 has crumpled structure, and has lamellar parts, microstructure of CGB can be tuned by the temperature, concentration, and N2 flow rate. That nanodegree of crumpling can be observed by DLS and TEM.

Reviewer 3 Report
Comments and Suggestions for Authors
This is an interesting paper concerning the production of a novel graphene oxide material and an examination of its potential application as a lubricant. I have some comments about the manuscript
1) The introduction contains a lot of studies related to other applications of crumpled graphene balls such as electrodes, and sensors. These applications are not directly linked to the study so it is not necessary to include so much details about each reference. I would suggest including more studies related to the use of graphene and related materials as lubricants and why the target material would be expected to function in this respect.
2) Please include the details of all the material utilized in this study. For example what as the purity of water used?
3) How was the aqueous dispersion of GO formed. Was the material sonicated? For how long, under what power? Was there any other preparation used? Please specify.
4) In line 144 you refer to "the same amount"of GO used in preparation of dispersion stability experiments. Please specify the quantity of GO used in each case.
5) The labels on the SEM pictures in figure 3 are very hard to read please alter the images to be easier to read.
6) Please include full descriptions in the captions for figures 2 and 3 like that for figure 4.
7) For the analysis of SEM and TEM images in figures 3 and 4 please include size distribution graphs to demonstrate any changes.
8) In figure 7 the internal graphs are labelled in Chinese, please change to English.
9) For a better comparison with the original material after friction testing please present images in figure 11 at the same scale as figure 4
Comments on the Quality of English LanguageThe English in the manuscript is generally good. There is a couple of strange phrases that need to be better written but I had no issue understanding the language.
Author Response
Reply to the reviewers’ comments and revision details
Title: The preparations of crumpled graphene oxide balls and research in tribological properties
Dear editor and reviewers,
Thanks a lot for your patient and constructive comments. We have carefully revised our manuscript by considering the reviewer’s comments. A point-by-point reply to the reviewer’s comments and revision details are as follows:
This is an interesting paper concerning the production of a novel graphene oxide material and an examination of its potential application as a lubricant. I have some comments about the manuscript.
1) The introduction contains a lot of studies related to other applications of crumpled graphene balls such as electrodes, and sensors. These applications are not directly linked to the study so it is not necessary to include so much details about each reference. I would suggest including more studies related to the use of graphene and related materials as lubricants and why the target material would be expected to function in this respect.
Response:
Introduction paragraph has been re-edited. It contains four parts. (1) The reason for using ionic liquids (with previous examples) (2) GO and CGBs were previously used to improve tribological properties of some lubricants (mainly oils) (3) Presentation on the methods for obtaining the desired CGBs. (4) A short paragraph on other uses of CGBs.
2) Please include the details of all the material utilized in this study. For example, what as the purity of water used?
Response:
All the material utilized in this study has been described in the 2.1 Material section. The corresponding modification is on page 3 of the article.
The water used in this experiment is ultrapure water. Millipore Teflon filter was purchased from Longjin Membrane Technology Co., Ltd., diameter: 60 mm, pore size: 0.22 μm. Commercially available steel balls (AISI-52100) with 6 mm diameter and steel substrate were used for friction tests. The steel balls and substrate were ultrasonically cleaned in pure alcohol for each test. Other materials were used as received.
3) How was the aqueous dispersion of GO formed. Was the material sonicated? For how long, under what power? Was there any other preparation used? Please specify.
Response:
The same amount (10 mg) of GO, CGB-1, and CGB-2 were added to three 5ml glass bottles respectively, and then 4ml water was added to obtain the water dispersion system of the sample. The dispersion stability of different systems was investigated by ultrasonic dispersion for 30 min, sonicated power at 160 W. There is no other preparation. The corresponding modification is on page 4 of the article.
4) In line 144 you refer to "the same amount" of GO used in preparation of dispersion stability experiments. Please specify the quantity of GO used in each case.
Response:
The same amount (10 mg) of GO, CGB-1, and CGB-2 were added to three 5ml glass bottles respectively, and then 4ml water was added to obtain the water dispersion system of the sample. The corresponding modification is on page 4 of the article.
5) The labels on the SEM pictures in figure 3 are very hard to read please alter the images to be easier to read.
Response:
Figure 3 has been improved, in addition, we also label the slice part of CGB-1 in Figure 3(a).
Figure 3. SEM images of CGB. (a) CGB-1 (b) partial enlargement of (a), (c) CGB-2 (d) partial enlargement of (c).
6) Please include full descriptions in the captions for figures 2 and 3 like that for figure 4.
Response:
The unclear captions of figures are re-edited, the modifications are as follows:
Figure 2. SEM and TEM images of GO. (a) SEM image of GO, (b) TEM image of GO, (c) corresponding SAED pattern of graphene oxide nanosheets. The inset (b) is percentage values of carbon and oxygen.
Figure 3. SEM images of CGB. (a) CGB-1 (b) partial enlargement of (a), (c) CGB-2 (d) partial enlargement of (c).
7) For the analysis of SEM and TEM images in figures 3 and 4 please include size distribution graphs to demonstrate any changes.
Response:
The sizes of the nano-particles were tested by DLS, as shown in Figure 5(b). With longer heating time, the average size of the CGB became smaller. The average size of CGB-1 is 495 nm, and the average size of CGB-2 is 586 nm. Although the two particles are similar, it can be seen from Figure 5(b) that the distribution range of CGB-1 is wider than that of CGB-2, indicating that the longer the heating time, most of the GO shrinks with N2 flow. The corresponding modification is on page 7 of the article.
Figure 5. (a) Raman spectra of GO and CGB; (b) the size distribution of the GO and CGB nano-additives.
8) In figure 7 the internal graphs are labelled in Chinese, please change to English.
Response:
Please forgive my carelessness, the Fig.7 has been modified as required. The following picture is the modified Figure 7. The corresponding modification is on page 9 of the article.
Figure 7. Friction coefficient-time curves of different concentrations of water dispersions and ionic liquid dispersions with different graphene additives.
9) For a better comparison with the original material after friction testing please present images in figure 11 at the same scale as figure 4.
Response:
TEM images before friction test has been added in Figure 12. The corresponding modification is on page 13 of the article.
Fig. 11 TEM images of GO, CGB and the debris after friction test: (a) GO; (b) GO after friction testing; (c) CGB-1; (d) CGB-1 after friction testing; (e) CGB-2; (f) CGB-2 after friction testing.

Round 2
Reviewer 1 Report
Comments and Suggestions for Authors
the comments raised previously have been addressed and the revised version of the manuscript is suggested for publication.
Author Response
Comments and Suggestions for Authors:the comments raised previously have been addressed and the revised version of the manuscript is suggested for publication.
If there is no other modification?
Reviewer 2 Report
Comments and Suggestions for Authors
Dear Sir,
The authors have satisfactorily answered to my queries, with one exception: which degree of crumpling is the best? The authors observed that the average size are 636 nm for GO, CGB-1 is approximately 495 nm, and CGB-2 is approximately 586 nm. Their conclusion upon comparing the two CGOs was that CGB-1 showed superior properties.
Simple GO with a size of 636 nm (with poor properties) is crumpled to 586 nm (which is not very good) and to 495 nm (which is good). Therefore, the question that I have asked if what if a CGO with intermediate dimensions (e.g. ca. 540 nm) or even smaller size (e.g. 460 nm) would be even better?
Can the authors, using their device, aim for a specific dimension by, as they say in their answer, tuning the temperature, concentration, and N2 flow rate? I suggest adding to the Conclusion paragraph a 3rd point with such a short discussion upon the future of this study, aiming to determine the most suitable dimensions, respectively the degree of crumpling, in function of the operating conditions (mainly temperature, concentration, and N2 flow rate).
Author Response
Dear editor and reviewers,
Thanks a lot for your patient and constructive comments. We have carefully revised our manuscript by considering the reviewer’s comments. A point-by-point reply to the reviewer’s comments and revision details are as follows:
(1) The authors have satisfactorily answered to my queries, with one exception: which degree of crumpling is the best? The authors observed that the average size are 636 nm for GO, CGB-1 is approximately 495 nm, and CGB-2 is approximately 586 nm. Their conclusion upon comparing the two CGOs was that CGB-1 showed superior properties.
Response:
Please forgive my carelessness, there is something wrong in the description of Fig.5. The average size of CGB-1 is approximately 586 nm, and the average size of CGB-2 is approximately 495 nm. It has been modified as required. The corresponding modification is on page 6 of the article. Highlighted in red in Revised Manuscript.
(2) Simple GO with a size of 636 nm (with poor properties) is crumpled to 495 nm (which is not very good) and to 586 nm (which is good). Therefore, the question that I have asked if what if a CGO with intermediate dimensions (e.g. ca. 540 nm) or even smaller size (e.g. 460 nm) would be even better?
Response:
According to the results of friction test, CGB with intermediate dimensions maybe better. CGB-1 has crumpled structure, also has lamellar parts. The part that wrinkled CGB-1 can avoid stacking after long-term friction, and lamellar parts formed a protective layer between the counterpart. The corresponding modification is on page 14 of the article. Highlighted in red in Revised Manuscript.
(3) Can the authors, using their device, aim for a specific dimension by, as they say in their answer, tuning the temperature, concentration, and N2 flow rate? I suggest adding to the Conclusion paragraph a 3rd point with such a short discussion upon the future of this study, aiming to determine the most suitable dimensions, respectively the degree of crumpling, in function of the operating conditions (mainly temperature, concentration, and N2 flow rate).
Response:
Thank you for your kindly recommendation, future research plan has been added in Conclusions section. The corresponding modification is on page 14 of the article. Highlighted in red in Revised Manuscript.
- To obtain the best lubricants with improved friction-reduction and wear-resistance properties in various lubricating states, fine-tuning of the degree of crumpling, in function of the operating conditions (mainly temperature, concentration, and N2 flow rate), would be investigated in our future work.
